# School-Based Intervention on Cardiorespiratory Fitness in Brazilian Students: A Nonrandomized Controlled Trial

**DOI:** 10.3390/jfmk4010010

**Published:** 2019-01-21

**Authors:** Giseli Minatto, Edio Luiz Petroski, Kelly Samara da Silva, Michael J. Duncan

**Affiliations:** 1Research Centre in Physical Activity and Health, School of Sports, Department of Physical Education Federal University of Santa Catarina. Campus Universitário – Trindade, Florianópolis, SC 88040-900, Brazil; 2Research Centre for Kineantropometry and Human Performan, School of Sports, Department of Physical Education Federal University of Santa Catarina. Campus Universitário - Trindade. Florianópolis, SC 88040-900, Brazil; 3School of Life Sciences, Faculty of Health and Life Sciences, Coventry University. James Starley Building. Priory Street, Coventry, CV1 5FB, UK

**Keywords:** physical fitness, children, adolescents, intervention study, physical education, school health, motor activity

## Abstract

Background: In response to the worldwide increasing prevalence of low cardiorespiratory fitness (CRF), several interventions have been developed. The aim of this study was to examine the effect of a school-based intervention on CRF in Brazilian students. Methods: A nonrandomised controlled design tested 432 students (intervention group: *n* = 247) from 6th to 9th grade recruited from two public secondary schools in Florianopolis, in 2015. The intervention entitled “*MEXA-SE*” (move yourself), applied over 13 weeks, included four components: (1) increases in physical activity during Physical Education classes; (2) active recess; (3) educational sessions; and (4) educational materials. CRF (20-m shuttle run test) was the primary outcome. Results: The effect size of the intervention on CRF was 0.15 (CI 95% = –0.04; 0.34). In the within-group comparisons, VO2max decreased significantly from baseline to follow-up in the control group but remained constant in the intervention group. After adjustment variables, differences between intervention and control group were not statistically significant (*p* > 0.05). Conclusion: The “*MEXA-SE*” intervention did not have an effect on adolescents’ CRF. However, maintenance of VO2max in intervention group and a reduction within control group demonstrates that this intervention may be beneficial for long-term CRF and, possibly, the increased intervention time could result in a better effect.

## 1. Introduction

Cardiorespiratory fitness (CRF) is considered an important marker of health in childhood and adolescence [1]. Low CRF has been associated with increased cardiovascular disease risk in young people [2] and adults [3]. Similarly, the maintenance of adequate CRF has been considered a protective factor for reducing the burden of mortality from cardiovascular diseases [4].

In response to the worldwide increasing prevalence of low CRF, several interventions have been developed and evaluated in recent decades [5,6]. However, the evidence of the effectiveness of school-based interventions for the promotion of CRF in low-and middle-income countries (LMICs) is limited [6,7]. Interventions implemented in the school environment prioritising CRF suggest that, regardless of study design and of the test used to measure CRF, the largest effects on CRF were found as a result of greater session length (>60 min), frequency of three weekly sessions, programmes lasting from 13 to 24 weeks [6], and higher-intensity physical activity (PA) [5]. The literature also points to high-intensity interval training (HIIT) as a feasible and time-efficient approach for improving CRF in adolescent populations [8]. However, the embedding of HIIT within the school day (e.g., in physical education or activities adapted for the classroom) is limited [8]. 

In terms of intervention strategies administered within the school day, systematic reviews suggest that engagement in PA is one of the major strategies to improve CRF, with the school being a favourable environment for this [5,6]. The methods used to improve CRF via PA include increasing the number and intensity of physical education (PE) lessons a week; inclusion of aerobic and resistance exercises; increasing PA within and outside school [6]; and a combination of printed educational materials and changes to the school curriculum [5]. Interventions which include changes to both PE classes and another aspect of school provision (e.g., strategies of exercise/PA (effect size = 0.88; CI 95% 0.55; 1.24), PA after school time (effect size = 0.44; CI 95% 0.00; 0.87)) have reported a greater effect compared to solely PE-based strategies (effect size = 0.14; CI 95% −0.03; 0.31) [6]. In this context, we applied an intervention based on the evidence previously cited, adapted to the context of schools in Southern Brazil, a low- and middle-income country (LMIC). Thus, the present study aimed to examine the effect of a school-based intervention on CRF among Brazilian students.

## 2. Materials and Methods 

### 2.1. Design and Participants

The “MEXA-SE” intervention was a nonrandomised controlled trial conducted in two (one experimental) secondary major schools of the South region of Brazil. Umbrella research had the objective of analysing the effect of a multicomponent intervention, applied during one school semester (approximately four months), on health-related physical fitness and body image of students from 6th to 9th grade. A detailed description of the full trial protocol can be accessed in the trial registration (Available online: https://www.clinicaltrials.gov/ct2/show/NCT02719704?term=NCT02719704&rank=1). 

According to the records of the Municipal Department of Education in 2015, 7484 students from 6th to 9th grade were enrolled in 26 public schools. The sample size calculation considered the following parameters: Effect size (ES) for each outcome, power of 80%, and significance level of 5% (Appendix A). Specifically, the calculation of the sample size for CRF required 35 people within each group, with an effect size of 0.68 [6]. To account for a potential 30% loss at follow-up, 46 participants per school were necessary. Following these parameters, the sample was calculated for all primary outcomes of the umbrella research. The largest sample size required among all outcomes was 295 students (see Appendix A), and this was the minimum sample established for the recruitment of schools.

School recruitment was based on the identification of the larger sample size (*n* = 295) and on the agreement of PE teachers in the intervention school to participate in the intervention. Of the 26 existing schools identified, five were eligible, two located in the Southern region (about 689 students) and three in the North (about 1165 students). For this study, the two schools (one control and one experimental) were selected in the same region of the city (Northern region) in order to reduce the possible socioeconomic disparities among students (mean of total monthly income of the people responsible for the house of each census tract of schools in Northern region = R$273,321 (approximately $73,473; €64,160) and Southern region = R$707,416) (approximately $190,166; €166,060)) [9]. Of the selected schools, one refused to participate and the third school from the same region was invited to participate. Contact with these schools was initiated in December 2014. The allocation of schools in the intervention (IG) and control (CG) groups was determined by authorities of the Municipal Secretariat of Education and the researchers had no influence in this decision. All participating schools were located within the urban perimeter of Florianopolis and most of the students resided near the school.

All students from grade 6th to 9th of these schools were eligible (*n* = 1,011) (records of the Municipal Department of Education). The final sample size adopted was 295 students (see Appendix A); however, for ethical reasons and at the request of the principals of each school, the intervention was conducted with all students in the schools. In the intervention school, all students could participate in the activities offered in the “*MEXA-SE*” (move yourself) programme. In both the intervention and the control school, only the students who delivered parental permission were evaluated. 

### 2.2. Intervention

#### 2.2.1. Theoretical Aspects

The intervention strategies were developed considering previous evidence obtained via systematic review with meta-analysis [6] prepared for this purpose. The meta-analysis variables considered were CRF (primary outcome); intervention setting (school only); and strategies in intervention (actions in PE classes and one other) and control (traditional PE classes) groups. Additionally, the type of exercise for the IG (aerobic and resistance) was considered, along with session duration (minimum 45 min), weekly frequency (three times), and duration of intervention (13 weeks or more). 

Intervention strategies (Table 1) related to PA were developed according to the theories of health promoting schools [10], sociocognitive theory [11], and the ecological model of health promotion [12]. Body image intervention was based on the sociocognitive theory [11] and the health belief model [13], and nutritional intervention was based on the dialogical model of health education [14]. The logical model of intervention (Appendix A) was developed in accordance with the suggestions of the Center for Disease Control and Prevention [15]. The logic model includes inputs available, developed activities, outputs, influencing factors, outcomes expected in the short and long term, and the desired aim of the intervention. 

#### 2.2.2. Intervention Strategies

The intervention employed in the current study had four integrated components that were delivered during school time in a PE class or other module. 

##### First Component: PE Classes

The first component was to increase time spent on in moderate to vigorous PA (MVPA) during the three PE lessons per week. The three weekly classes were conducted by PE teachers in the school. The lessons were composed of approximately 10 min of stretching exercises, 10 min of strength exercises/muscular endurance, and 20 min of aerobic exercise, prioritising activities that arouse students’ interest and in which most of them were involved in movement. The content was organised according to the Curriculum Proposal for PE of Florianopolis [16] and the National Curriculum Parameters of PE [17]. Thus, we used different content from PE (games, sports, dancing, martial arts, gymnastics) and prioritised playful aspects of learning. In total, students received an average of 25 (SD = 6.4) PE lessons (45 min per lesson). 

##### Second Component: Active Recess

Students were also encouraged to increase PA practice during school recess. Volleyball, basketball, football, futsal, handball, and ropes were available for students to occupy the school recess time actively. These materials could be used on the school environment, such as sports courts.

##### Third Component: Educational Sessions

Educational sessions on “Health, Lifestyle, Physical Activity and Sedentary Behaviour”: These sessions were planned by researchers and conducted by school PE teachers and lasted 45 min each. The first session aimed to discuss issues related to health and healthy lifestyle, seeking to identify beneficial and harmful health behaviours. The second session was aimed at discussing PA, physical exercise, and sedentary behaviour, seeking to identify the physical activities practised by students, clarifying concepts, demonstrating the importance of each behaviour for health, and reflecting on changes that everyone could do to become more active. For the development of sessions, video, educational games, and posters were used.

Educational sessions on nutrition: These sessions consisted of six sessions (45 min per session) designed to promote reflection and positive changes in eating habits and healthcare (to improve knowledge and eating habits), conducted by a nutritionist with each school year group separately. The topics developed in the sessions were: (1) Healthy eating; (2) general recommendations about the choice of foods in terms of natural and processed meals; (3) consuming a wide variety of organically grown fruit and vegetables; (4) guidelines on how to combine foods in a meal; (5) guidelines on the act of eating; and (6) a cookery workshop. The teaching methods used were movies, expository lectures, workshop context posters, music, and cooking workshops. 

Educational sessions on body image: These sessions comprised three sessions (45 min per session) focused on body image satisfaction conducted by a PE researcher. The topics developed in the sessions were: (1) Beauty standards; (2) individual qualities; and (3) preparation of a poster on the theme: "What is beauty to you?". The teaching methods used were movies, expository lectures, and workshop context posters. The intervention was delivered to each class (*n* = 18) separately.

##### Fourth Component: Education Materials

Leaflets about sedentary behaviour and PA outcomes were distributed at the school, and two folders were sent to the parents (Appendix A) by students. The first folder had messages about reducing sedentary behaviour (recommendations, tips for changing of this behaviour, etc.), and the second on increasing PA (the importance of parental incentive, examples and benefits of active living for youths). The folders were also given by the PE teacher to the students with messages specific (Appendix A), along with educational sessions on these behaviours (third and fifth weeks).

School PE teachers participated in a training programme (4 hours of duration) for the implementation of intervention strategies. Training consisted of an exhibition about the current context of health-related physical fitness and body image in adolescence, group dynamics that integrated physical fitness components in a practical way, and from the presentation of the intervention proposal, the role of the PE teacher was highlighted in the intervention and support material was delivered. In addition to training conducted with PE teachers, all school teachers participated in a 90-minute meeting to present the aims and activities of the “*MEXA-SE*” programme.

##### Control Group

Both the IG and CG received the standard school curriculum as determined by the Brazilian government, which allocates 135 min of PE classes per week (3 school sessions). The mandatory PE curriculum was the content of games, sports, dancing, martial arts, and gymnastics. The school activities of the CG remained unchanged. The three weekly 45-min PE classes were conducted by PE teachers at school, following annual planning. 

### 2.3. Variable Measures

CRF, the primary outcome, was assessed using the 20-m SR test using standard procedures [18], validated for Brazilian use [19]. We analysed the following indicators of 20-m SR: Laps; stages; minutes; and maximum oxygen consumption (VO2max), using the equation proposed by Leger et al [18]. 

Anthropometric measurements were conducted by anthropometrists certified by The International Society for the Advancement of Kinanthropometry [20]. Calculations of technical error of measurement were carried out for all anthropometric measures, which are considered acceptable for experienced evaluators [21] (height: Intra evaluator = 0.28%, inter evaluator = 0.20%; triceps skinfold (TR): Intra evaluator = 1.64%, inter evaluator = 3.91%; and subscapular skinfold (SE): Intra evaluator = 2.64%, inter evaluator = 7.27%). 

Overall PA was self-reported using a list of MVPA validated for Brazilian adolescents (ICC = 0.88; CI 95%: 0.84–0.91) [22] and showed a reproducibility Kappa value of 0.45 (89.3% agreement). This list [22] included PA which was organised into different PA types: Collective PA/sports (e.g., soccer, basketball, volleyball, and indoor soccer, in 7 items), individual PA/sports (e.g., swimming, athletics, martial arts, and gymnastic, in 8 items), ride in PA (e.g., skateboarding/rollerblading and cycling, in 2 items), walking (i.e., leisure, transportation, and walking with dog, in 3 items), popular games (e.g., dodgeball and “forty-forty”, in 2 items), and strengthening PA (e.g., weight training and abdominal exercises, in 2 items). Students reported the frequency and duration of each daily PA that they performed in the previous week. Thus, we estimated the weekly time (in minutes) of total MVPA. 

The time spent in MVPA within school was measured with an Actigraph GT3X+ accelerometer, secured on the right hip by an elastic band around the waist. Wear time was determined by subtracting the time when the accelerometer was given to children (beginning of class) from the time the accelerometer was retrieved (end of class). Students wore the device for four days (from Monday to Thursday) during school time (from 7:30 to 11:30 or from 13:00 to 17:00). We considered valid data using the accelerometer for three or more days and for at least three hours per day. Data were analysed in 15-second epochs [23]. The measure used in this study to characterise the PA at school was the total minutes of MVPA, according to Everson et al [23] cut points.

Socioeconomic status (SES) was measured using a questionnaire [24]. This questionnaire estimates the purchasing power of families and ranks them from richest (A1, A2, B1, B2) to poorest (C1, C2, D, E) based on the accumulation of material goods, housing conditions, number of working individuals in the household, and the education level of the household head. The instrument provides a score for each item according to the amount present in the home and the degree of instruction of the head of the household. Finally, the sum of the scores obtained in all the items makes it possible to classify according to the abovementioned classes (Available online: http://www.abep.org/criterio-brasil). For purposes of analysis, classes A1, A2, B1, B2 were grouped into “A+B”, and C1, C2, D, E were grouped into “C+D+E”. 

Sexual maturation was self-assessed by the participants by classifying their breast (girls) and genital (boys) development in five pubertal stages, as proposed by Tanner [25] and validated at the UFSC (Kendall’s correlation coefficient of 0.627 (*p* < 0.01) for boys and 0.739 (*p* < 0.01) for girls) [26]. The students were classified as prepubescent (stage 1), pubescent (stages 2 to 4), and postpubescent (stage 5).

Attendance in PE lessons was registered by the PE teacher in each lesson. This procedure was conducted before the start of activities. 

### 2.4. Data Collection

The data collection team was composed of professors and students of the undergraduate and graduate PE and nutrition courses. Team members received training for the application of questionnaires and for the standardisation of measurements and motor tests. The instruments were applied in the respective order: Questionnaire (first day), anthropometric measurements, 20-m SR test, and sexual maturation (second day). Data collection was performed during students’ class period at school. The average duration was 10 days at baseline and at follow-up (Figure 1). It was not possible to blind the staff as to which group the assessed students belonged to because the availability of human resources was limited. It was also not possible to blind students and PE teachers to the allocation due to the intervention characteristics (different activities to those conducted in PE classes before the start of the intervention). The intervention timeline is shown in Figure 1. 

### 2.5. Statistical Analysis

Means and standard deviations were calculated at baseline and post-test for continuous variables (CRF, age, body mass, height, sum of TR and SE skinfold (TR+SE), PA in school, and overall PA). The normality of data was determined by values of skewness and kurtosis (±3) [27], confirmed by the display of values and histograms. The height and differences between post- and pre-test for minutes, laps, stages and VO2max showed normal distribution. The variables of body mass, TR+SE, minutes, and laps were transformed by log, VO2max by reciprocal (1/VO2max), and MVPA within school by square root. Chronological age, minutes of practice of overall PA, and stages of sexual maturation did not present normal distribution. 

Mean and proportions differences between intervention and control participants at baseline were compared by independent *t* (parametric variables) and Mann–Whitney *U* (nonparametric variables) tests and chi-square analysis, respectively. To determine the effects of the intervention, analysis of covariance (ANCOVA) was used, with the change in CRF as the dependent variable, groups as the independent variable, and baseline data of CRF, sex, age group, SES, sexual maturation, overall PA, MVPA in school, and TR+SE as the covariates. The effect of the intervention was also tested from the intention-to-treat analysis (dropout data imputed by the repetition of the last observation—simple imputation) using analysis of covariance in order to assess the possible impact of sample losses in the intervention effect. In addition, we tested the interaction of all covariates with the intervention effect. We considered the existence of interaction when the *p*-value < 0.10 [28]. The level of significance for the study was 5% for two-tailed tests using the statistical software SPSS 15.0® (SPSS IBM Inc., Chicago, USA). The effect of the intervention on CRF was calculated for each outcome using the standardised mean difference (SMD) with a 95% confidence interval (95% CI) in Review Manager.

The analyses performed had a statistical power higher than 80% and 5% significance level for two-tailed tests. In the adjusted analysis of variance with group vs. time (baseline and follow-up comparisons, considering a conservative intermeasured correlation of 0.1 (Available online: http://www.gpower.hhu.de/), the ES found was ≥0.10. 

### 2.6. Ethical Considerations

This study was conducted in accordance with the Declaration of Helsinki and it was approved by the Ethics Committee on Human Research of Carmela Dutra Maternity (process No. 780.303). The informed written consent of parents or guardians and the assent of participants were obtained. 

## 3. Results

### 3.1. Participants

Of 1854 students enrolled in the five biggest schools, 1011 (two schools: 568 in IG and 443 in CG) from 6th to 9th grade (aged 9 to 16 years) were invited to take part. Of these, 568 students provided parental consent and personal assent (60.2% in IG and 51.0% CG). Of 568 participants, 89.8% (IG) and 93.4% (CG) completed baseline measures. In follow-up measures, the response rate was 80.5% and 87.7% in IG and CG, respectively. Considering the time of intervention, the reasons for dropout were absence (10.7%), giving up (7.1%), motor limitation (3.2%), and school change (2.5%). Finally, 432 students (247 in the IG, 185 in the CG) participated in the study at both baseline and follow-up (Figure 2).

#### Deviations

Deviations from the planned study delayed the onset of intervention by four weeks because of a delay on liberation of schools by the Municipal Department of Education; a teachers’ strike from the fifth to the eighth week (13 days of lessons) of intervention also disrupted the schedule. After the delay, the intervention occurred in 11 weeks (Figure 2).

### 3.2. Comparison of Baseline Characteristics

In baseline measurements, students in the CG had greater mean values for body mass, TR+SE, attendance to PE lessons, and low performance in the CRF test than the IG (*p* < 0.05). Baseline data were also compared between dropouts and students who completed the intervention. Dropouts had greater mean values of body mass and TR+SE, and fewer minutes of MVPA practice, and less attendance of PE lessons. However, there were no differences for age, height, CRF (minutes, laps, stages, and VO2max), sex, SES, and sexual maturation variables (Table 2).

### 3.3. Efect of Intervention

The effect size of intervention was 0.15 (CI 95% = −0.04; 0.34). According of CRF indicators (Table 3), the effect size for all indicators was low and not significant (CI_minutes_ 95%: –0.03; 0.35, CI_laps_ 95%: −0.06; 0.32, CI_stages_ 95%: −0.08; 0.30, and CI_VO2max_ 95%: −0.06; 0.33). In all CRF indicators, no significant differences were observed for the IG between baseline and follow-up. For the CG, only VO2max decreased significantly from baseline to follow-up. The results for adjusted differences between the IG and CG and intention-to-treat analysis were not significant. There was no interaction between the groups and sex, age, sexual maturation, and SES.

## 4. Discussion

The results of the current study show that the “*MEXA-SE*” programme had a small but nonsignificant effect on CRF among students. We found a significant reduction in VO_2_max from baseline to follow up in the CG, while no significant alteration was seen in the IG. Other indicators of CRF analysed (minutes, laps, stages, and VO2max for IG) did not differ statistically between or within groups. These results are similar to results from a cluster randomised controlled study [29] that evaluated the effect of an intervention targeting the physical and organisational school environment for noncurricular PA (SPACE) on CRF in Danish adolescents (11–14 years, mean: 6m; CI 95%: −20; 31, *p*-value: 0.43). 

Conversely, the findings of the current study are contrary to results from a nonrandomised controlled trial [30] conducted during school time in Brazilian students (10 to 15 years old). Intervention strategies were applied in PE classes twice a week for 60 min for one school year; the structure of PE classes comprised 30 min of aerobic exercise, 20 min of playing sports, and 10 min of stretching activities. The results for the nine-minute test were significant for CRF between groups (ES = 0.30; CI 95% = 0.10; 0.50) [30]. The long duration of sessions (60 min) and the length of intervention (one school year) [30] can explain the differences in effect size between studies with the same design (nonrandomised controlled trial) realised in Brazil. Although the weekly frequency of the study conducted in the same country (twice a week) [30] was lower than that of the present study (three times a week), the longer duration of intervention helped to overcome the resistance of the students to the new exercises. 

Another reason for our results may be the duration of intervention. The intervention was planned to last 14 weeks, in line with recommendations for promoting change in this outcome [6]. However, this was not possible due to the delay of the start of the intervention (four weeks) and the teachers’ strike that stopped lessons for 13 days of the intervention (from the fifth to the eighth week). The teachers’ strike in Brazil also interrupted other interventions conducted with students [31]. This external factor is a reality present in Brazilian schools and is a further obstacle in promoting an improvement in physical fitness components. Researchers working in environments such as that found in Brazil should be alert to teachers’ strike conditions for future interventions and might be able to mitigate the effect of such issues by planning longer interventions and/or considering the possibility of this type of interruptions. However, we do not consider that this interruption due to the teachers’ strike had a meaningful impact on the results of the intervention, as similar results (no significant ES) were found in other multicomponent interventions of a longer duration [32,33].

One of the effective strategies to improve CRF highlighted in literature is the inclusion of aerobic and resistance exercises [6], and higher intensity PA in PE classes [5]. These strategies were included in the “*MEXA-SE*” programme. However, the fact that there is one "free lesson" per week would make it impossible to determine that this strategy is not effective. The PE teacher in the previous year used the “free lesson” system (students can do what they want during class, including staying seated) for all lessons, and with a new class structure, many students were resistant to participation; one of the three PE lessons continued in the “free lesson” system by agreement with students. 

In the current study, of the 45 min total in the lesson, the PE teacher needed 15 min to record the presence and outline general content procedure before commuting to the sports field, leaving only 30 min to apply other content. The meta-analysis that aimed to determine the effectiveness of interventions designed to increase the proportion of PE lesson time that students spend in MVPA showed, on average, a 24% relative increase in the amount of lesson time spent in MVPA [34]. The Center for Disease Control and Prevention (CDC) [35] has previously recommended that students should be engaged in MVPA for at least 50% of PE lesson time. This information suggests that pedagogical practice of the “*MEXA-SE*” programme was inefficient and could be improved. 

Other intervention designs have shown positive effects on the CRF promotion, particularly those using HIIT [8]. Meta-analysis regarding the utility of HIIT to improve CRF in adolescents evidenced the little heterogeneity (Q = 9.77, I^2^ = 28.3%, *p* = 0.202), and a large ES (*d* = 1.05, CI 95% 0.36 to 1.75). Ten min of HIIT training has shown to have comparable results to 40 min of moderate aerobic training [8]. Although the evidence of embedding HIIT within the school day is limited, this type of intervention has the potential to improve CRF in adolescent populations [8,36]. 

The findings from the CG identify that standard PE lessons and other activities did not contribute to improving the CRF of Brazilian students, and a reduction in VO2max was identified in the present study. In addition, attendance in PE lessons was higher for the CG compared to the IG. These results confirm the hypothesis that standard school activities, including PE lessons, do not add sufficient resources to promote and/or maintain CRF in students. Investigations into PE lessons in Brazil showed the reality of exercise in schools, i.e., the reduced mean duration of the lessons (35.6 min, SD = 6.0) and low mean proportion of time spent in MVPA (32.7%, SD = 25.2 or 12.3 min, SD = 9.7) [37]. This directly affects one of main settings to contribute to improvement of physical fitness [6], because for organic adaptations to occur as a result of PA, individuals should be subjected to moderate and/or intense efforts taking place over a certain period [36]. Considering that the only difference between the IG and CG was the exposure to the intervention, another possible explanation for these results is the reduction in CRF that naturally happens with advancing age [38].

There are, of course, some limitations in this study. Firstly, the design was nonrandomised, because school assignment to the intervention group was made intentionally by the local educational authorities. Secondly, we were unable to conduct the study ‘double-blind’ because the intervention consisted of such a combination of obvious changes and activities that the data collection staff (as well as the students and PE teachers) would certainly be aware of which person or school was in which study condition. Thirdly, the intervention started four weeks later than anticipated and during the intervention period, there was a teachers’ strike (May 2015), which resulted in a gap in intervention activities of 13 days. As a consequence, the duration of intervention was 11 weeks rather than the initially planned 14 weeks. Finally, the response rate for post-testing at the IG was lower than expected due to the number of absences of students from school, possibly because data collection took place in the final week of the school semester. 

This study was based on the available evidence about interventions on CRF in adolescents around the world [6]. Consequently, it contributes to the advancement of interventions on CRF in LMICs, as Brazil.

## 5. Conclusions

In conclusion, the “*MEXA-SE*” programme contributed to the maintenance of CRF, compared to a CG where CRF declined in the same period. On the other hand, the lack of changes in the school environment and maintenance of PE classes in the usual model cannot help to maintain this component. The results need to be interpreted with caution due to extraneous factors (e.g., delayed onset of intervention, teachers’ strike, and free lessons) that occurred during intervention. This intervention should be retested to show better the real effect on the CRF of the students. 

## Figures and Tables

**Figure 1 jfmk-04-00010-f001:**
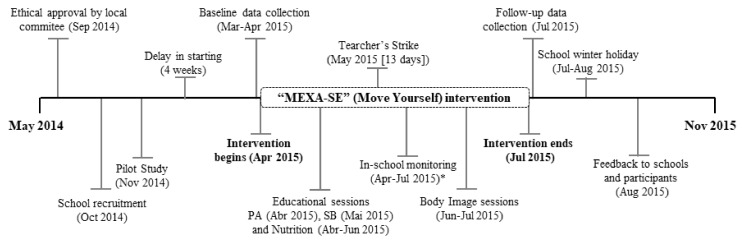
Timeline of the “*MEXA-SE*” intervention study. Note: PA: Physical activity; SB: Sedentary behaviour; *In-school monitoring comprised observations of physical education classes and school recess.

**Figure 2 jfmk-04-00010-f002:**
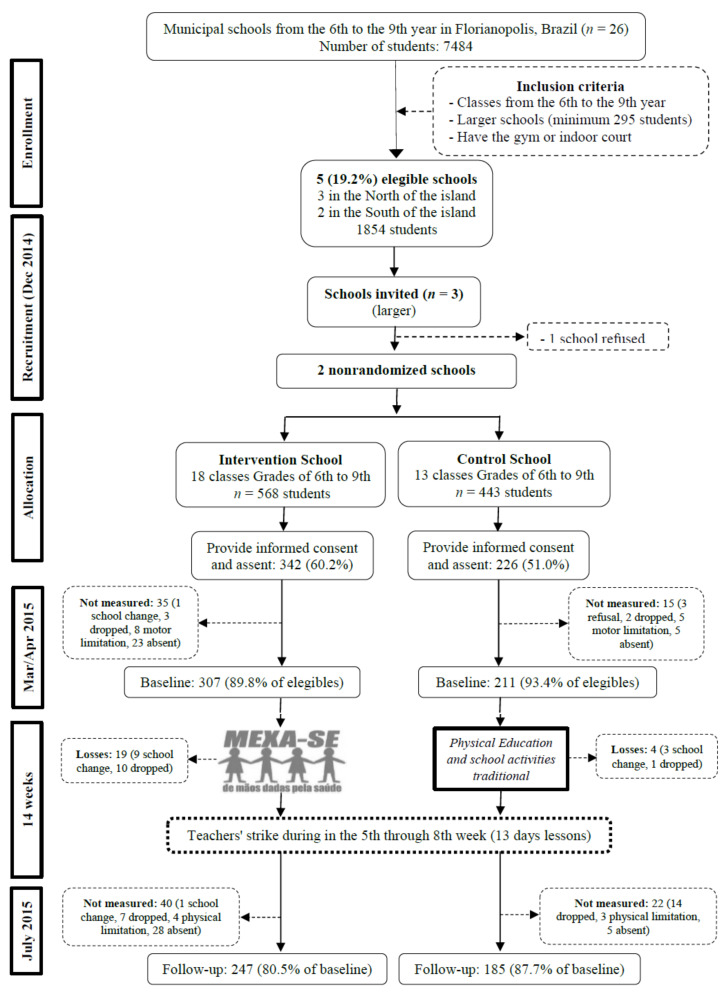
Flowchart of cardiorespiratory fitness. Florianopolis, Brazil, 2015. Mar: March; Apr: April; Dec: December.

**Table 1 jfmk-04-00010-t001:** Description of “*MEXA-SE*” actions.

Actions	Influence Level	Theory	No. Sessions	Duration	WF	Executing Agent
Training for PE teachers	School	EPS	1	4 h	-	Researchers (PE)
PE Classes: Stretching exercises (10 min), strength and muscular resistance (10 min), and increased intensity in the main part of the class (MVPA)	Individual	Meta-analysis	42 *	45 min	3	School PE teachers
Active recess	Individual, School	EPS; TSC; Meta-analysis	70 *	15 min	5	Researchers and School
Educational sessions on the following topics: Health, lifestyle, physical activity, and sedentary behaviour	Individual	EPS; TSC	2	45 min	†	School PE teachers
Educational sessions on healthy eating and nutrition	Individual	MDES	6	100 min	1	Researchers (Nutritionist)
Educational sessions related to body image	Individual	TSC; Belief in health	4	45 min	1	Researchers (PE)
Placement of posters about physical activity and sedentary behaviour in school and health units	School	EPS; MSE	-	-	†	Researchers, students and teachers
Distribution of leaflets on physical activity and sedentary behaviour	Individual, Family	EPS; MSE	-	-	††	School PE teachers Researchers

PE: Physical education; WF: Weekly frequency; MVPA: Moderate to vigorous physical activity; EPS: Health-Promoting Schools [10]; Meta-analysis [6]; TSC: Sociocognitive theory [11]; MSE: Ecological social model [12]; MDES: Dialogical model of health education [14]; * We considered 14 weeks of intervention; † fixed in the 3rd and 5th week of the intervention; †† delivered to students on the 3rd and 5th week of intervention and to parents on the 4th and 6th week.

**Table 2 jfmk-04-00010-t002:** Baseline physical and sociodemographic characteristics, and physiological measures in intervention and control groups in Brazilian students. Florianopolis, SC, Brazil, 2015.

Variables	IG (*n* = 247)	CG (*n* = 185)	*p*-value	All (*n* = 432)	Dropout (*n* = 136)	*p*-value
	Mean (sd)	Mean (sd)		Mean (sd)	Mean (sd)	
Age (years) †	12.4 (1.3)	12.7 (1.3)	0.052	12.6 (1.3)	12.8 (1.4)	0.084
Body mass (kg) *	47.7 (11.3)	50.2 (12.4)	0.047	48.8 (11.8)	52.9 (13.4)	0.002
Height (cm) *	155.5 (9.8)	156.6 (9.9)	0.237	156.0 (9.8)	157.8 (9.8)	0.082
TR+SE (mm) *	22.5 (11.5)	24.4 (11.5)	0.042	23.3 (11.6)	26.9 (13.3)	0.004
MVPA school (min)	10.8 (6.6)	10.4 (6.6)	0.499	10.8 (6.6)	10.1 (5.6)	0.833
PA general (min) †	688.6 (913.2)	648.2 (831.3)	0.935	671.0 (877.7)	572.4 (902.6)	0.029
Attendance PE lessons (%) †	83.8 (13.3)	92.0 (8.3)	<0.001	87.3 (12.2)	72.4 (23.4)	<0.001
Minutes *	3.4 (1.8)	3.0 (1.7)	0.009	3.2 (1.8)	2.9 (1.5)	0.093
Laps *	26.4 (16)	22.8 (13.9)	0.013	24.8 (15.2)	21.9 (12.3)	0.077
Stages †	3.8 (1.9)	3.3 (1.7)	0.007	3.6 (1.8)	3.2 (1.5)	0.134
VO2max (mL/(kg·min) *	41.7 (4.8)	40.0 (4.6)	<0.001	41.0 (4.8)	40.0 (4.5)	0.075
	% (*n*)	% (*n*)	*p*-value **	% (*n*)	% (*n*)	*p*-value **
Sex			0.769			0.833
Male	47.4 (117)	45.9 (85)		46.8 (202)	47.8(65)	
Female	52.6 (130)	54.1 (100)		53.2 (230)	52.2 (71)\	
Socioeconomic Status			0.190			0.385
A + B	54.0 (129)	47.5 (86)		51.2 (215)	46.7 (57)	
C + D + E	46.0 (110)	52.5 (95)		48.8 (205)	53.3 (65)	
Sexual maturation††			0.705			0.067
Prepubescent (S1)	1.7 (4)	1.1 (2)		1.4 (6)	1.0 (1)	
Pubescent (S2 to S4)	86.8 (210)	89.4 (160)		87.9 (370)	79.8 (79)	
Postpubescent (S5)	11.6 (28)	9.5 (17)		10.7 (45)	19.2 (19)	

IG: Intervention group; CG: Control group; sd: Standard deviation; TR+SE: Sum of triceps and subscapular skinfold; PA: Physical activity; MVPA: Moderate to vigorous physical activity measured with an accelerometer; m: Metres; kg: Kilogram; mm: Millimetres; VO_2_max: Maximum oxygen consumption. S: Stages; † Mann–Whitney test; †† Breasts and genitals; * Independent *t* student test and ** qui-square test.

**Table 3 jfmk-04-00010-t003:** Effect of “*MEXA-SE*” intervention on cardiorespiratory fitness (CRF) among Brazilian students. Florianopolis, SC, Brazil, 2015.

	Differences between Baseline and Follow-Up				Adjusted Differences between Intervention vs. Control Group					Interaction			
Indicator CRF	Intervention (*n* = 247)	*p*-value	Control (*n* = 185)	*p*-value	Adjusted Difference (*n* = 432)	*p*-value	Intention-To-Treat Analysis (*n* = 518)	*p*-value	ES	Group vs. Sex	Group vs. Age	Group vs. SM	Group vs. SES
	Mean (CI 95%)		Mean (CI 95%)		Mean (CI 95%)		Mean (CI 95%)			*p*-value	*p*-value	*p*-value	*p*-value
Minutes	0.05 (−0.10; 0.21)	0.517	−0.11 (−0.28; 0.06)	0.206	0.17 (−0.07; 0.40)	0.173	0.13 (−0.09; 0.34)	0.257	0.13	0.351	0.194	0.558	0.949
Laps	0.67 (−0.58; 1.91)	0.224	−0.83 (−2.24; 0.57)	0.291	1.50 (−0.46; 3.46)	0.134	1.16 (−0.63; 2.95)	0.202	0.16	0.477	0.179	0.759	0.963
Stages	0.07 (−0.09; 0.22)	0.308	−0.06 (−0.23; 0.11)	0.497	0.13 (−0.12; 0.37)	0.307	0.09 (−0.13; 0.31)	0.430	0.11	0.122	0.293	0.894	0.724
VO_2_max	−0.27 (−0.72; 0.18)	0.240	−0.75 (−1.24; −0.24)	0.004	0.48 (−0.23; 1.19)	0.186	0.40 (−0.26; 1.05)	0.232	0.14	0.107	0.494	0.900	0.882

CI: Confidence interval; VO2max: Maximum oxygen consumption; EF: Effect size adjusted; SM: Sexual maturation; SES: Socioeconomic status; *p*-value of ANCOVA analysis adjusted by baseline CRF, sex, age group, sum of triceps and subscapular skinfold, sexual maturation, socioeconomic status, physical activity general, time in moderate to vigorous physical activity, and percentage of attendance in PE lessons (Adjustment for multiple comparisons: Bonferroni).

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
