# Peer review of "School-Based Intervention on Cardiorespiratory Fitness in Brazilian Students: A Nonrandomized Controlled Trial"

_jfmk, 2019, doi:10.3390/jfmk4010010_

Round 1

Reviewer 1 Report

With great pleasure I have read  the  manuscript dedicated to an important aspect  of the lifestyle reconditioning . Some minor aspects seem to  be clarifyed . 

 The first consideration is on the CRF that  is  the  principal  aim of the  study.  . The  age of the population investigated is too low. The students were adolescent or preadolescent and in anycase in the age where the potential modification on terms of VO2 max  are unexpected .

Level of instruction of the family  and of  the students .

Did you try  to differentiate by gender the   eventual differences of the values  obtained ?

 Where the schools were located in city or not? How far the students distance from the school?

 Did you measure  the spontaneous PE? This  is  the basic level for  the  aerobic performance . In case of low level of spontaneous PA  the values  of CRF are inevitably  low. 

Author Response

- With great pleasure I have read  the  manuscript dedicated to an important aspect  of the lifestyle reconditioning . Some minor aspects seem to  be clarifyed . 

Answer: Thank you.

- The first consideration is on the CRF that  is  the  principal  aim of the  study.  The  age of the population investigated is too low. The students were adolescent or preadolescent and in anycase in the age where the potential modification on terms of VO2 max  are unexpected .

Answer: Yes, we agreed. We chose this age group to intervene because the literature shows that the earlier the improvement in cardiorespiratory fitness is stimulated, especially in Physical Education classes, the greater the chances that the benefits will be experienced even at an early age. As in this age group the potential modification on terms of VO2 max  are unexpected, the results observed are consistent with the literature.

- Level of instruction of the family  and of  the students.

Answer: The level of instruction of the family is contemplated in the indicator of socioeconomic status and the level of instruction of the students is from 6th to 9th grade, according to information presented in the abstract, method (Design and participants), and results (participants – text and flowchart).

- Did you try  to differentiate by gender the eventual differences of the values  obtained?

Answer: Yes. The results of the moderators of the intervention effect are being addressed in another manuscript under evaluation.

- Where the schools were located in city or not? How far the students distance from the school?

Answer: Yes, the schools were located in city. This information was add in method section. We do not have precise information on the distance from the home to the school of each student, but most of them resided in the school district itself.

- Did you measure  the spontaneous PE? This  is  the basic level for  the  aerobic performance . In case of low level of spontaneous PA  the values  of CRF are inevitably  low. 

Answer: We agreed, but unfortunately we do not have this information.

Reviewer 2 Report

This manuscript is on a non-randomised study in Brazil. The intervention was multicomponent and contained several components. However, the major limitation for the study is the low number of schools involved. It is not certain what is the purpose of the study and what the authors aim to achieve through such a study. Was it a feasibility trial? if so, then the authors may like to rename the study as a feasibility and provide details of the feasibility. The language could be improved and here are some comments for the authors to consider.

--

Abstract

There are too many acronyms in the abstract, I suggest removing ig cg, pe,

L28, include 'the' before within-group

L29, did you mean to replace "considering the" with "after"?

Intro

There is a lack of scientific references. Improving the style by avoiding the use of ref 6 in consecutive sentences. L. 46-50

I would like to suggest that the authors provide more information about the different types of crf tests, the superior tests, and how they may be used. Some practical information would be good to include. 

L64-66. The aim of the study is not clear for me.

Materials

The link for clinical trials does not work. Include a correct one. 

Although the instruments are listed in the trials documentation, it does not provide actual protocol of the study. It would be good to know who administer the tests, in what order, and how accurate the reporting is. 

L81-82, it is not clear what is meant. If the number was supposed to be 46, then the experiment is over sampled, what bias were controlled for? 

L83-85, it is not clear how the schools were identified. 

L87, what do you mean by island? 

L89, could you also put the currency in USD and EUR? 

L94-5, it is it clear how the numbers were created. 

L128, It is not clear if there were 3 pe classes per week, or 3 pe classes in the intervention. 

L133, what is meant by physical culture?

L142, there needs to be clearer information about the educational sessions. Please provide a table with when the sessions occurred in relation to each other to accompany figure 1.

L203, it is not clear what a list of mvpa is. Write explicitly the measurement tools, especially when they are self report.

L209-10, it is not clear if the device was worn at the beginning of the class or at the school day?

L215, state explicitly the ses instrument and its use for analyses. 

L219, state explicitly the tanner instrument and how the instrument was used in the analyses.

L240, provide reference for skewness +/-3.

L251-4, it is not clear what the imputation method was, and why that was the case, when the subsequent line states the use of only data with valid data.

L255, it is not clear what is meant by <. 10

Results

L271-275, it is difficult to follow when the numbers 568 appears more than once. It would be good to include the actual numbers next to the percentage.

L289, include actual p value when less than. 05, but if less that. 001, then use the less than symbol.

L302, why is the data not shown?

Discussion

L333-340, is speculative hence remove. No data about the the notices were carried out.

L361, include "the" before current study

L363, include "the" before meta-analysis

Study limitation is the lack of fidelity and process evaluation is much needed.

Author Response

This manuscript is on a non-randomised study in Brazil. The intervention was multicomponent and contained several components. However, the major limitation for the study is the low number of schools involved. It is not certain what is the purpose of the study and what the authors aim to achieve through such a study. Was it a feasibility trial? if so, then the authors may like to rename the study as a feasibility and provide details of the feasibility. The language could be improved and here are some comments for the authors to consider.

Answer: thank you for comments. The study is not feasibility trial. The native English coauthor (M.D.) revised the language. Thank you so much for your contribuitions.

Abstract

There are too many acronyms in the abstract, I suggest removing ig cg, pe,

Answer: the acronyms “PA, IG, CG, and PE” were removed. The changes are highlighted in text.

L28, include 'the' before within-group

Answer: this word was included.

L29, did you mean to replace "considering the" with "after"?

Answer: yes, we changed  the “consideding the” to “after”.

Intro

There is a lack of scientific references. Improving the style by avoiding the use of ref 6 in consecutive sentences. L. 46-50

Answer: reference was added in this section, and the use of ref 6 in consecutive sentences was avoided.

I would like to suggest that the authors provide more information about the different types of crf tests, the superior tests, and how they may be used. Some practical information would be good to include. 

Answer: There is no evidence with practical information about CRF testing and its relation to the effect found in interventions in adolescents. The results of the meta-analysis cited in the introduction (ref 6) showed inconclusive results. This information is in the introduction, on lines 47 and 48 (“Interventions implemented in the school environment prioritizing CRF suggest that, regardless of study design and of the test used to measure CRF, the largest effects on CRF were found...”).

L64-66. The aim of the study is not clear for me.

Answer: the aim of the study was adjusted.

Materials

The link for clinical trials does not work. Include a correct one. 

Answer: thank you. The correct link was included.

Although the instruments are listed in the trials documentation, it does not provide actual protocol of the study. It would be good to know who administer the tests, in what order, and how accurate the reporting is. 

Answer: this information was added in the manuscript.

L81-82, it is not clear what is meant. If the number was supposed to be 46, then the experiment is over sampled, what bias were controlled for? 

Answer: this information has been better clarified in text.

L83-85, it is not clear how the schools were identified. 

Answer: this information has been better clarified in text.

L87, what do you mean by island? 

Answer: Florianopolis is a island. We changed to “city” in text.

L89, could you also put the currency in USD and EUR? 

Answer: Yes, done.

L94-5, it is it clear how the numbers were created. 

Answer: this information has been better clarified in text.

L128, It is not clear if there were 3 pe classes per week, or 3 pe classes in the intervention. 

Answer: were 3 PE classes per week and the intervention was applied in this classes. This information has been better clarified in text.

L133, what is meant by physical culture?

Answer: this term is not correct. Thank you for asked us. We changed in text.

L142, there needs to be clearer information about the educational sessions. Please provide a table with when the sessions occurred in relation to each other to accompany figure 1.

Answer: we do not have the precise information of when each session happened for each class, only the period as shown in Figure 1. All classes received the sessions in the established period. Table 1 gives more information about the educational sessions that complement Figure 1.

L203, it is not clear what a list of mvpa is. Write explicitly the measurement tools, especially when they are self report.

Answer: this information was added.

L209-10, it is not clear if the device was worn at the beginning of the class or at the school day?

Answer: this information has been better clarified in text.

L215, state explicitly the ses instrument and its use for analyses. 

Answer: more informations were added in text.

L219, state explicitly the tanner instrument and how the instrument was used in the analyses.

Answer: more informations were added in text.

L240, provide reference for skewness +/-3.

Answer: a reference was added.

L251-4, it is not clear what the imputation method was, and why that was the case, when the subsequent line states the use of only data with valid data.

Answer: this information was added and adjustments in the description were done in text.

L255, it is not clear what is meant by <. 10

Answer: this information has been better clarified in text.

Results

L271-275, it is difficult to follow when the numbers 568 appears more than once. It would be good to include the actual numbers next to the percentage.

Answer: we opted to keep only the percentage in the text because both absolute values and percentage are plotted in Figure 2. If we present both values we will be repeating the information that is in the figure. To avoid duplicity, we kept it as it is.

L289, include actual p value when less than. 05, but if less that. 001, then use the less than symbol.

Answer: the specific p-value is in Table 2, as indicated at the end of the paragraph. We chose to use in the text "p <.05" to avoid duplication of information.

L302, why is the data not shown?

Answer: in fact the data is shown, are those presented in the text only, not in the table. We remove the information "(data not shown)".

Discussion

L333-340, is speculative hence remove. No data about the the notices were carried out.

Answer: these lines were removed.

L361, include "the" before current study

Answer: done.

L363, include "the" before meta-analysis

Answer: done.

Study limitation is the lack of fidelity and process evaluation is much needed.

Answer: we faced some difficulties throughout the intervention period. The lack of fidelity pointed out by the reviewer occurred, however, possible measures to mitigate the impact of this were taken. The process evaluation of the intervention is in progress.